# Are Psychophysiological Wearables Suitable for Comparing Pedagogical Teaching Approaches?

**DOI:** 10.3390/s22155704

**Published:** 2022-07-30

**Authors:** Vesna Geršak, Tina Giber, Gregor Geršak, Jerneja Pavlin

**Affiliations:** 1Faculty of Education, University of Ljubljana, 1000 Ljubljana, Slovenia; vesna.gersak@pef.uni-lj.si (V.G.); tina.giber@gmail.com (T.G.); jerneja.pavlin@pef.uni-lj.si (J.P.); 2Faculty of Electrical Engineering, University of Ljubljana, 1000 Ljubljana, Slovenia

**Keywords:** wearables, teaching approaches, embodied cognition, distance teaching, synchronization, knowledge gain

## Abstract

This study describes how wearable devices can be used in elementary schools to compare some aspects of different teaching approaches. Upper arm wearables were used as an objective tool to compare three approaches when teaching science: (i) classical frontal teaching, (ii) embodied (kinesthetic) teaching, and (iii) a distance teaching approach. Using the wearables, the approaches were compared in terms of their impact on students’ psychological arousal and perceived well-being. In addition, short-term and long-term knowledge gain and physiological synchronization between teacher and students during the lecture were assessed. A synchronization index was defined to estimate the degree of physiological synchronization. During distance teaching, by means of measurements with wearables, students were significantly less physically active and significantly less psychologically aroused. Embodied teaching allowed significantly higher physical activation than during the other two approaches. The synchronization index for all three teaching approaches was positive with the highest values for distance and frontal teaching. Moreover, knowledge gain immediately after the embodied lessons was higher than after frontal lessons. No significant differences in the long-term knowledge retention between the three different teaching methods were found. This pilot study proved that wearables are a useful tool in research in the field of education and have the potential to contribute to a deeper understanding of the mechanisms involved in learning, even in complex environments such as an elementary school classroom.

## 1. Introduction

One of the challenges of education today from kindergarten to university is to reduce students’ sedentary time and incorporate more physical activities into the curriculum in order to improve concepts such as learning outcomes, student well-being, learning effectiveness, and long-term retention [1,2]. During the 2020 coronavirus pandemic, increased sedentariness was combined with increased screen time at home [3], and the overall effects on students were even more alarming.

The integration of physical activity into teaching and learning attributes the increased effectiveness of learning partly to the increased number of memory channels (visual, kinesthetic and tactile) [4] and partly to the underlying neurophysiological principles, i.e., physical activity increases the growth of neurons and synapses [5,6,7]; assists the long-term memory [8,9]; increases integration of cognitive, motor, and sensory information between cerebral hemispheres [10]; increases oxygen flow due to increased heart rate [11]; and so on [2]. However, these facts do not focus on the psychophysiological responses of students during physical activity, so new learning models based on the latest neurological findings are still expected. Based on these findings, the so-called embodied (kinesthetic) teaching approach was developed.

Embodied teaching and learning is a contemporary teaching approach in which the human sensory system is activated by delivering new, additional, and/or redundant information through audiovisual, olfactory, tactile, and kinesthetic channels [12]. Cognitive science uses embodied cognition theory to describe the influence of the human motor system on cognition [12,13,14,15,16,17,18,19,20,21]. Embodied teaching as a teaching approach enables students to use movement to express, form, and create various educational content [22], and its benefits are proven in various science fields, such as chemistry [23,24], mathematics [25,26,27], natural science and engineering [28], and elementary education [29,30]. In the field of human–computer interaction, the embodied teaching and learning have not been extensively studied, but currently there is an emerging body of research related to it (see review in [31]). Different physiological parameters, e.g., indices derived from children’s electrodermal activity, heart rate, heart-rate variability, and EEGs, are monitored using various measuring devices. The physiology is studied with pedagogical and social measures in order to assist explanation of the states and traits and the behavior of the schoolchildren and their teachers during the learning process [32,33,34,35]. In general, studies indicate that physiological data of schoolchildren (collected in ecologically valid media) can correlate to their learning experience [36].

### Wearables in Education

Among the tools that help researchers to discover processes and activities within learning (and teaching) when placed in a real environment are wearable devices. These can monitor various parameters of the observed students (and teachers) during the process of teaching and learning in a complex physical and sociological environment such as a classroom. Autonomous wearable measurement devices that are worn on the user’s body to monitor the user’s physical, (psycho)physiological, mental, emotional, and behavioral activity are currently being used with increasing regularity [36,37,38,39,40,41,42,43,44,45,46,47,48,49,50,51] but are not widely used in empirical research in educational science. They are usually used to measure students’ physical activity, and few are used to measure students’ psychophysiology [35,52,53].

Today’s wearable technology is progressing rapidly but still has certain limitations (data collection rate, limited autonomy, health and ergonomic issues, lack of validation of its measurement function [38,52], dependence on environmental conditions). However, wearables work well in real classroom conditions [36,53]. Wearables have the potential, unlike psychological measurement instruments such as knowledge tests and questionnaires, to be a reliable, objective research tool that could provide new psychological, cognitive, motor, and sociological information about the neuropsychological aspects of learning and teaching [54,55,56,57]. They can even be used for detection of students’ emotions induced by their environment, classmates, teacher, and the teaching itself [36,45,58,59,60]. According to [36] the use of wearables enables students and their teachers to monitor and reflect on the learning process in addition to studying complex learning experiences in more efficient and accurate ways.

One of the aims of this study was to create a tool for estimating the physiological synchronicity of persons involved in the process of teaching and learning. The physiological connection between two persons has been extensively studied in other environments [61]. The phenomenon of physiological connection/synchronization/concordance is observed in the fields of education, philosophy, psychology, psychotherapy, neuroscience, etc. [62,63,64,65,66,67]. It was shown that synchronization can be connected to mirroring processes underlying empathy or the ability to share and understand others’ emotional and mental states and shared attention or even used as the measure of intensity of gaming [66,68,69]. In educational science the physiological synchronization, if well-defined and commonly accepted, could in principle be used as an indicator of those effective time intervals during teaching, which results in a more effective and deeper/enhanced learning (e.g., resulting in better knowledge gain).

The main goal of this paper is to question the use of wearable devices in classroom and in addition to present a study using wearable devices in a real-world setting. A class of elementary school students were taught certain topics in science using three different teaching approaches. Using wearables, we were able to assess the differences in teaching approaches, psychological arousal, and general feeling of the students and the teacher. In addition, by monitoring physiology of students and the teacher, the physiological connection between them could be estimated. The aim was also to identify science knowledge gain.

The paper is structured as follows: After the Introduction, in which an extensive literature review is outlined in order to identify the role of wearable devices in educational science and to present different teaching approaches, the following Materials and Methods section describes participants, measurement instruments, study protocol, and data processing. The following Results section consists of the main findings and their interpretation and is followed by a comprehensive Discussion. The paper is rounded off by a brief Conclusion.

## 2. Materials and Methods

### 2.1. Participants

A class of 25 11-year-old fifth graders participated in the study. In addition to the students, three adults were involved—one active teacher and two assistants. Out of 25, four students were formally classified as special needs students with various learning disabilities (e.g., attention deficit, attention deficit hyperactivity disorder (ADHD), graphomotor problems).

All participants wore a wearable measuring device throughout the pedagogical experiment. Due to absence of students, measurement errors, physical detachment of the wearable devices, or excessive motion artefacts, not all of the 25 students contributed to the total acquired data (i.e., in four teaching sessions only data of 20, 18, 22, and 22 students could be processed (Figure 1)).

### 2.2. Instrumentation

The physiological data were acquired using a wearable device. For estimation of emotional feeling of the student, the student’s emotional (well-being) questionnaire was administered.

Physiological data were acquired by a wearable device, BodyBug by Bodimedia Inc. (Figure 2, left). Attached to the subject’s upper arm by elastic bands, it is a light, autonomous device with internal memory capacity which uses dry stainless-steel electrodes to measure several physical and physiological parameters. The reliability and accuracy of the device was thoroughly evaluated in previous studies [40,44,59,68,69,70,71,72,73].

For the purpose of this study the following was measured: skin temperature, electrodermal activity (EDA, skin conductance level (SCL), and skin conductance response (SCR)) and heat flux of the upper arm, and energy expenditure and estimated physical activity of the student (metabolic equivalent of task (MET)) [74]. Before the experiment, in order to accurately measure energy expenditure, each wearable was adjusted to the body characteristics (age, height, weight, and handedness) of the student wearing it.

The BodyBug wearable proved to be an ergonomically very suitable wearable to explore children’s activity at school, as it was perceived as non-intrusive by the majority of involved students (Figure 2, right) [57].

To avoid health issues and minimize the personal characteristics of students, each wearable was associated with a dedicated student throughout the study.

The student emotional feeling questionnaire was designed to assess students’ well-being. It was designed as a 5-point Likert smiley faces scale [75]. On the scale from 1 (very discontent) to 5 (very content), the students marked the smiley which described their current emotional feeling the best. One of the reasons this questionnaire was included is the lack of valence component in the used psychophysiological measures. i.e., EDA is a measure of arousal level but not the valence (the subject can be very excited, but their positive or negative emotional content is unknown). Thus, the well-being scale was used to estimate the students’ emotional valence.

To determine science knowledge, the knowledge tests created specifically for the purpose of the study, taking into account existing science and technology subject curricula [76] and commonly identified misconceptions, were used three times as a pre-test, post-test 1 (immediately after instruction), and post-test 2 (after 4 months to determine long-term effects). The knowledge tests were always the same for each lesson; only the questions in the test were mixed so that the results were comparable by lesson. All knowledge tests, except the distance learning test, were in physical (paper) form. After the distance learning test, the knowledge test was administered in online form.

### 2.3. Study Protocol

Students in a fifth grade class in an urban elementary school were taught various topics in the subject science and technology using three different types of teaching approaches.

To partially avoid common subjective biases in teacher–student relationships during classroom work, students were taught by a new teacher. The new teacher met with the students prior to the experiment to establish a certain bond and trust relationship. During the experiment, the new teacher met the students four times. Each of the four sessions was 2 h long. Originally, only two approaches were planned, but during the study the coronavirus pandemic lockdown occurred, and distance teaching was included as the third teaching (and learning) approach in the study. In the end, three teaching approaches were compared: (i) classical sedentary ex cathedra frontal teaching, (ii) embodied (kinesthetic) teaching, and (iii) online distance sedentary teaching.

The study lasted for 2 months, with the long-term knowledge retention evaluation after 4 months.

At classes conducted physically in school, the wearables were attached to the upper arm of the students before the beginning. To assess synchronization effects, wearables were worn also by the teacher. In distance teaching classes, the wearables were shipped to students and their teacher to their homes. The student emotional questionnaire was filled in online after the session. All four sessions were video recorded. Note that in the case of distance teaching, only the students’ faces were visible. In addition, students received instructions on how to attach the sensors prior to the online lessons. The students were already familiar with the online application and, from the school lessons, familiar with self-attaching the sensors. Therefore, no help from parents was required for distance teaching (and thus their direct influence was avoided). During distance lessons, the teacher was instructed to design the lesson so that students moved with similar intensity to frontal work in school. Thus, the main potential limitations of online lessons were the quality of the Internet connection and distractions at home.

The topics were selected from the Slovenian national curriculum for science and technology [76,77,78]. To explore correlation of physiology with topic difficulty and to control for differences in topics difficulty, prior to this study, an expert panel (six experienced teachers of science) estimated topic difficulty on a 5-point Likert scale, 1 being the easiest and 5 the hardest topic (Table 1).

The procedure was approved by the Ethics Committee for Pedagogy Research of the Faculty of Education, University of Ljubljana, and consent forms were collected prior to the study from the schools’ administrations, teachers, and parents of the children.

### 2.4. Data Analysis

During the sessions, students’ skin temperature, EDA, energy expenditure, and level of physical activity were recorded. The raw data were transferred to a computer and reviewed using the SenseWear 8.3 software package (by Bodimedia Inc.). The measured data were coded by students and sessions and further processed in AcqKnowledge 5.0 (by Biopac Inc.) Microsoft Excel and RStudio software packages. The data were then analyzed according to the topics of an individual session, e.g., relation of physiology with difficulty of the topics, teaching approach or special needs status, correlations of selected measures, and synchronization of EDA. The processed data were additionally interpreted with observational data, a semi-structured interview with the teacher and videos of the meetings, but this is beyond the scope of this paper and is not presented here.

EDA data were processed by preconditioning the raw EDA data (manual filtering, outlier and moving artefact removal), and calculating the skin conductance levels. In addition, the number of skin conductance responses (SCRs) was determined using 0.05 Hz high pass filtering with a 0.02 microS response threshold [79,80,81].

To estimate the physiological synchronization between two signals, Pearson correlations were calculated in certain time windows. The rather low sampling frequency of the wearable (4 samples per minute) resulted in selecting the length of the time window of 1 min. Using the correlation coefficients, a synchronicity index was calculated as the ratio of positive coefficients and negative coefficients as suggested in [64,82]. Synchronization between different signals was calculated, e.g., between teacher and average student signals, between teacher and individual student signals, and between teacher and teacher assistant signals. Note that for the purpose of this paper, physiological synchronization refers to synchronization of the EDA. Thus, a positive value of the synchronicity index represented a stronger EDA link of the selected signals and negative value a weaker EDA link. The results were additionally interpreted from the video recordings of the sessions post festum.

Physiological measures were processed with several statistical tests. Data distribution normality was checked with the Shapiro–Wilk test of normality. The Wilcoxon test with Bonferroni corrections (*p* ≤ 0.05) was used for comparing physiological measures of all sessions. For comparing the two groups, children with disabilities and children without disabilities, we used the t-test. The EDA synchronization phenomenon was estimated using the Pearson correlation coefficient. We interpreted the direction (positive and negative) and strength of synchronization, defining 0–0.20 as no correlation, 0.20–0.40 as weak correlation, 0.40–0.70 as medium correlation, 0.70–0.85 as strong correlation, and above 0.85 as very strong correlation [83].

Science knowledge was evaluated by knowledge tests. Data were processed similarly using descriptive and inferential statistics [83]. To find the effectiveness of the teaching approaches, an average normalized gain g was calculated from the percentage of correct answers on the knowledge test before and after the implementation of the topic (Equation (1)).
(1)$$g=\frac{\mathrm{correct}\;\mathrm{answers}\;\mathrm{after}\;\mathrm{the}\;\mathrm{lesson}\left(\%\right)-\mathrm{correct}\;\mathrm{answers}\;\mathrm{before}\;\mathrm{the}\;\mathrm{lesson}\;\left(\%\right)}{100\%-\mathrm{correct}\;\mathrm{answers}\;\mathrm{before}\;\mathrm{the}\;\mathrm{lesson}\;\left(\%\right)}$$

Benchmarks to define low gain (0–0.3), medium gain (0.3–0.7), and high gain (0.7–1.0) are listed by [84]. To compare knowledge after 4 months (i.e., long-term knowledge retention) with knowledge immediately after instruction (i.e., short-term knowledge retention), we calculated the g factor, i.e., “sustainability of knowledge” [85].

## 3. Results

### 3.1. Physiology and Teaching Approach

In order to compare three different teaching approaches, wearables were used to measure skin temperature (ST), electrodermal activity (SCL and SCR), and heat flux (HF) of the upper arm and energy expenditure and estimated physical activity (MET) of each student and the teacher throughout four teaching sessions.

As expected, students were on average more physically active during the embodied teaching and quite inactive during distance teaching. Significant differences were found in both measures of physical activity: energy expenditure (frontal–embodied 1 *p* = 0.078; frontal–embodied 2 *p* = 0.0000458, p.adj. = 0.000275; frontal–distance *p* = 0.016 p.adj. = 0.094; embodied 1–embodied 2 *p* = 0.035, p.adj. = 0.212; embodied 1–distance *p* = 0.00061, p.adj. = 0.004; embodied 2–distance *p* = 0.0000153, p.adj. = 0.0000918, Figure 3) and MET (frontal–embodied 1 *p* = 0.078; frontal–embodied 2 *p* = 0.0000763, p.adj. = 0.000458; frontal–distance *p* = 0.023 p.adj. = 0.139; embodied 1–embodied 2 *p* = 0,07, p.adj. = 0.042; embodied–distance *p* = 0.000427, p.adj. = 0,003; embodied 2–distance *p* = 0.000214, p.adj. = 0.001, Figure 4).

The embodied 1 session included movements, e.g., students were showing, presenting, and interpreting the science phenomena using their body, but during the embodied 2 session students were even more physically active, i.e., were actively moving, dancing, and jumping, hence the energy expenditure difference in Figure 3.

From the results of the students’ electrodermal activity, it can be concluded that all three teaching approaches elicited similar psychological arousal in the students, which means that the teacher managed to engage the students’ interest in learning and their active engagement in the lesson equally for each session. The higher SCR in the embodied 2 session can be attributed to the higher psychological arousal and mental involvement during the more physically active of the two embodied sessions (frontal–embodied 1 *p* = 0.529; frontal–embodied 2 *p* = 0.0005, p.adj. = 0.031; frontal–distance *p* = 0.145, p.adj. = 0.87; embodied 1–embodied 2 *p* = 0.038, p.adj. = 0.228; embodied 1–distance *p* = 0.009, p.adj. = 0.055; embodied 2–distance *p* = 0.000454, p.adj. = 0.003, Figure 5).

In addition, it was found that the average value of the students’ body heat flux (HF) held important new information. Students were statistically more psychologically activated (aroused) on average, i.e., had higher levels of HF, during distance teaching, while the other two approaches showed similar activation (frontal–embodied 1 *p* = 0.358; frontal–embodied 2 *p* = 0.611; frontal–distance *p* = 0.003, p.adj. = 0.016; embodied 1–embodied 2 *p* = 0.389; embodied 1–distance *p* = 0.000305, p.adj. = 0.002; embodied 2–distance *p* =0.003, p.adj. = 0.016, Figure 6). Research [86] found that arousal during testing correlated with students’ grades, i.e., the more aroused students were, the higher their grades were. In our study, students were less aroused during distance teaching, which was also perceived to be the easiest session. Therefore, we suspect that the low EDA level is due to the low difficulty of the subject.

ST was not significantly different during sessions (ST(frontal) = 31.2 °C, SD = 1.04 °C; ST(embodied 1) = 30.6 °C, SD = 1.28 °C; ST(embodied 2) = 31.0 °C, SD = 1.08 °C; ST(distance) = 31.0 °C, SD = 1.09 °C). Similarly, the SCL during four sessions was comparable (SCL(frontal) = 0.439 μS, SD = 0.251 μS; SCL(embodied 1) = 0.418 μS, SD = 0.364 μS; SCL(embodied 2) = 0.526 μS, SD = 0.487 μS; SCL(distance) = 0.362 μS, SD = 0.268 μS)). This indicates that the general psychological arousal of the students was comparable for all the sessions, meaning the teacher taught the different topics with a similar success in engaging and motivating the students.

The results of the student emotional questionnaire for estimating students’ well-being during lessons show that the students felt good (Figure 7). Our conclusion is that the psychophysiological responses can be classified with positive valence.

The correlation of students’ physiology and the topic difficulty was estimated, and correlation was found with ST and body HF (Figure 8). No correlation was present with SCL and SCR measures.

By comparing the physiology of students with and without learning disabilities, we could not conclude that students were significantly different in terms of physiology (HF, ST, and EDA).

### 3.2. EDA Synchronization

In our study, the synchronization was based on linking two selected EDA signals—the teacher’s EDA was selected as the central signal and compared with other EDA signals, such as the average student, the student with learning disabilities, and the most successful student. Thus, by calculating correlations and then analyzing the video recording of the session, we were able to find events (of a few minutes in length) where the signals were strongly related. The synchronicity index was a useful measure indicating the degree of synchronization (negative values meaning no connection and positive a high connection). Positive synchrony, i.e., high connection between students and teacher, was noted when solving worksheets together (embodied teaching), when performing the topic kinesthetically together (embodied teaching), and when watching the teacher mime (distance teaching). In addition to strong positive, we noticed strong negative synchronization found in moments when students were not connected to the teacher and not focused on the activity, e.g., during distance teaching, the teacher had problems with the Internet connection, which led to a decreased interest of the students and a consequent decrease in the synchronization index to negative values.

Since EDA synchronization is a rather complex phenomenon that can be observed and interpreted in different ways, and our definition of the synchronization index proved to be a useful tool for assessing the connection between teacher and students. Comparing synchronization indices between teacher and students, we found that the synchronization index during embodied teaching was 0.69 and 0.61 (for two lessons), during frontal teaching 1.25, and during distance teaching 1.07. These results are consistent with the content of the sessions, i.e., students were more focused on the teacher in frontal and distance teaching, whereas when they worked in groups during embodied learning, they were focused more on each other and paid less attention to the teacher. The calculated synchronization indices proved this finding.

We also observed intersubjective synchrony between the teacher and student. Their EDA signals were most positively synchronized when the teacher and the student communicated directly, e.g., when a student was listening and the teacher asked a question and the student raised his hand, responded, and waited for the teacher’s answer and further explanation. On the other hand, the EDA signals between the teacher and the student were strongly negatively connected when the teacher explained while the student was doing something else, such as drawing, as can be seen in the video recording (Figure 9). Signals were strongly negatively associated during breaks when the teacher and student were not engaged in the same activities.

Table 2 shows the results related to knowledge gain. The students scored higher on both post-test 1 (immediately after the lesson with specific topic) and on post-test 2 (after 4 months).

The results of the one-way analysis of variance ANOVA show statistically significant differences between knowledge gain immediately after the lesson (g_1_) depending on the type of lesson (frontal, embodied, or distance) (F = 3.443; *p* = 0.037; η^2^ = 0.085); g-factors in knowledge gain are the highest for embodied lessons, similar to the g-factors after distance education and significantly higher than the average g-factors of students after frontal lessons without embodied teaching. Moreover, the descriptive g-factor statistics for sustainability of knowledge show negative average values of the g-factor, which could explain the effect of forgetting [87]. The minimum g_2_ was −3 and −2, respectively, which means that students lost 2 and 3 times as many points, depending on their maximum possible progress. We found that the g-factor of sustainability shows the greatest loss in the lesson on the water cycle, where knowledge progress was also greatest. The smallest knowledge loss is shown in the lesson on soil layers and pollution. However, the Kruskal–Wallis test (KW = 2.084; *p* = 0.353 > 0.05) shows that there are no significant differences in the sustainability of knowledge in terms of the g-factor between the three different teaching methods.

## 4. Discussion

The main finding of the study is that it is worthwhile to use wearables in the classroom to conduct research in the field of educational science, e.g., to optimize teaching to achieve learning objectives. A comparison of different pedagogical approaches using wearables recording physiology of students was proven to be possible. The results of our study showed that the following physiological measures of students differed significantly according to teaching approach: heat flux and SCR, energy expenditure, and physical activity (MET).

As expected, students spent the most energy during embodied teaching and the least during distance teaching. Overall, the data show that students were most engaged during embodied teaching, which is also reflected in medium knowledge gain, consistent with the results of [88]. The embodied teaching and the distance teaching exhibited maximal levels of HF, SCR, EE, and MET. In distance teaching, we measured the highest HF level (which can be interpreted as the lowest cognitive load or highest relaxation of students), the lowest EE and MET (i.e., physical inactivity of students), and the lowest SCR number (which means lower psychological activation or arousal of students).

The results show that the more difficult the topic, the lower the HF levels and average skin temperatures, and conversely, the easier the topic, the higher the ST and HF levels. This was statistically significant for HF. However, similar to [89,90] lower levels of HF and skin temperature may indicate higher cognitive load, but they may also be the result of an individual’s anxiety or psychological arousal. Nevertheless, the identified relationship between the student’s physiology (HF) and the difficulty of the subject would, in principle, allow the design of complex pedagogical tools such as an affective textbook, i.e., a textbook that automatically responds to the level of perceived difficulty and adapts its content so that the student gains more knowledge, improves their motivation, etc.

One of the questions of our pilot study was whether simple wearables are useful to detect complex phenomena, such as physiological synchronization of different persons in the classroom. The results show that synchronization between students and teachers can be detected to some extent using wearables. For example, using synchronization indices, we were able to explain that students were more focused on the teacher during individual work (frontal and distance teaching), and they were more focused on each other and paid less attention to the teacher during group work during embodied learning. Nevertheless, our general conclusion is that although the synchronization index proved useful, it is not sufficient by itself for a detailed analysis of the link between two physiological signals. It must be accompanied by a detailed (and time-consuming) interpretation of actual classroom activities, e.g., video recordings.

When comparing students with and without learning disabilities, our results revealed no significant differences, in contrast to previously reported lower EDA values for the ADHD students [91].

The importance of using the body in the learning process has been pointed out by many philosophers, psychologists, and educators for decades [92]. Today, the effectiveness of the embodied (kinesthetic) approach compared to traditional teaching methods and the currently ongoing and widespread distance teaching can be tested and interpreted in many ways. One way is through the use of psychophysiological measurements. Through observation and collection of additional qualitative data, they help us understand what is happening to the student in the classroom. Combining the student’s physiology measured by wearables with observation of the student’s psychophysiological activity, behavior, knowledge testing, and knowledge retention and self-assessment of well-being provides a comprehensive view of the student’s condition at a given point in time.

Using psychophysiological measurements, we found that teaching approaches differ from each other. We have shown that there is a relationship between physical and psychological activation. In embodied sessions where students moved more (high EE and high MET), students were more psychologically active, more aroused, had a higher SCL and a higher number of SCR, and even showed the greatest progress on knowledge tests (not described in this paper). Our results show that physical activity affects psychological arousal. The latter is reflected in better attention, memory, decision making, and consequently higher efficiency [86].

Our results showed significantly higher short-term knowledge retention for embodied lessons compared to frontal lessons. There was no statistical difference of knowledge gain for distance teaching and no significant differences in the sustainability of knowledge (long-term knowledge retention) between the three different teaching methods.

In addition to the limited number of subjects, an inherent limitation of the study is the mental, cognitive, as well as environmental complexity of a classroom. One limitation of the study was low sampling rate of the wearable devices and non-optimal attachment of the wearables to the upper arms of the students to ensure optimal electrical contact and motion artefact-free data collection. Occasionally, readjustment of the wearable was necessary, which increased potential measurement anxiety in students. Environmental conditions were monitored but not controlled, e.g., classroom air temperature and relative humidity, CO_2_ content, lighting, acoustic noise, time of day. In addition, personality traits, professional experience, and teaching style of the teacher for the selected science topics were not taken into account. All these sources of error have been identified and minimized to the best of our knowledge, but along with student characteristics such as fatigue, disinterest, and motivation, they remain to be explored in future research [93,94].

Other possible influences that could affect the emotional response of the students and thus represent a limitation of such a highly complex study are the new teacher and researcher present, but this was kept to a minimum since both had been present in the class several times before, and the students were already familiar with them. The class is a living system, so unplanned events such as another teacher arriving in the classroom are possible and not unexpected for the students.

The effects of the altered environment in distance teaching were found to be negligible. From the video observation of students’ reactions and from the responses in the student well-being questionnaires, it can be concluded that there were no significant changes in emotional reactions. For example, students did not show significant reactions to the teacher’s unexpectedly unstable Internet connection. Whether students focused more on the distance teaching itself or on the learning content during distance teaching was answered by the knowledge test results—focusing on the learning content was noticeable from better knowledge gain. Distance teaching and learning has its pitfalls, but because of the trust that the teacher built with the students, we believe that the negative effects were greatly reduced as students made an effort to participate in a manner similar to school.

Due to the size of the sample, the results from Slovenian schools cannot be generalized to the general population, but we expect similar results in other countries, as science curricula elsewhere are similar to those in Slovenia. The repeatability of the study also depends on the preparation of the teacher, but since usually only enthusiastic teachers are willing to cooperate in this type of research, we expect similar results.

## 5. Conclusions

In our study, wearables were used to evaluate pedagogical teaching approaches. Three different teaching approaches were compared in terms of psychological arousal and perceived well-being of the students and the teacher, as well as physiological synchronization between the two. It was found that embodied teaching was the most energy demanding and engaging for students. Students during distance teaching were significantly less physically active and significantly less psychologically aroused. We interpret this to be related to student interest, which is an important component of motivation in education and is defined as psychological arousal that includes focused attention, increased cognitive functioning, persistence, and emotional involvement. In addition, this pilot study presents some observations in the synchronization between the students and the teacher that could provide insights into learning and teaching for educational science.

In conclusion, our pilot study proved that inclusion of wearables in research in the field of education could contribute to a deeper understanding of the mechanisms involved in learning. It showed that wearables could be used in complex actions such as comparing teaching approaches, including those that use movement to encourage students to achieve learning goals.

## Figures and Tables

**Figure 1 sensors-22-05704-f001:**
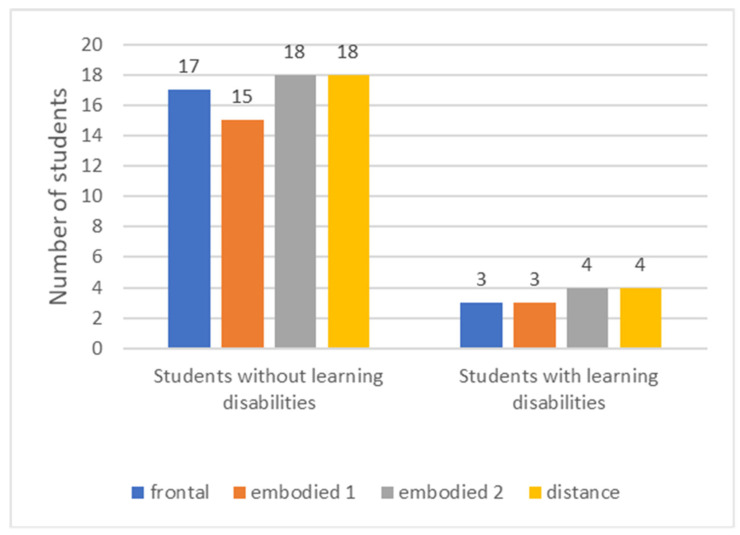
Number of students present at four teaching sessions (frontal, two embodied, and a distance session).

**Figure 2 sensors-22-05704-f002:**
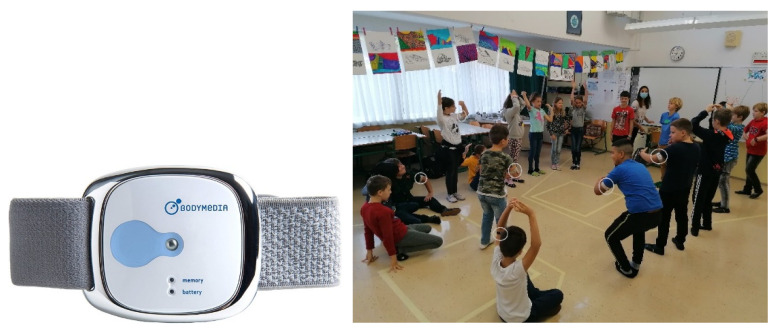
**Left**—upper-arm wearable device BodyBug Bodimedia. **Right**—teaching of science topics by means of the embodied teaching approach. Students are learning how the particles are distributed in matter. Wearables on students’ upper arms are visible (encircled).

**Figure 3 sensors-22-05704-f003:**
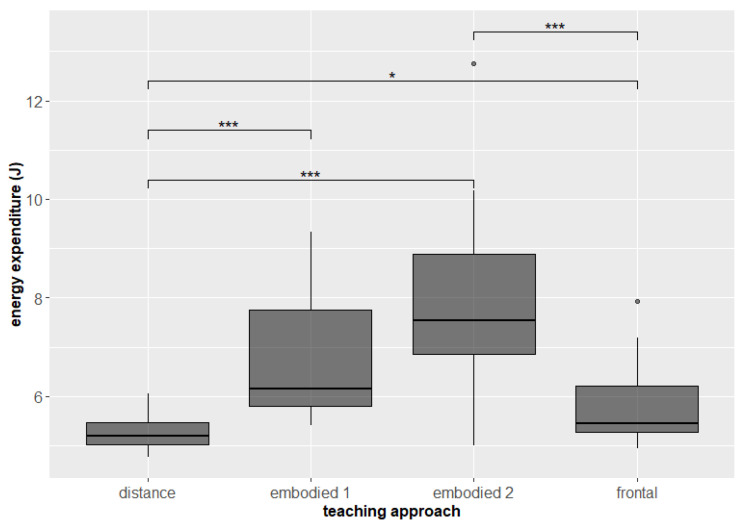
Energy expenditure of students during different teaching approaches. Significant differences are marked with * for *p* < 0.05, and *** for *p* < 0.001.

**Figure 4 sensors-22-05704-f004:**
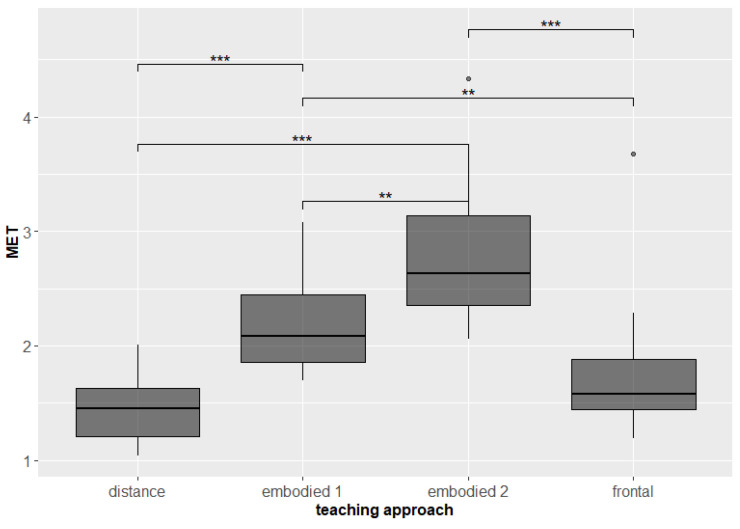
Metabolic equivalent of task (MET) of students during different teaching approaches. Significant differences are marked with ** for *p* < 0.005, and *** for *p* < 0.001.

**Figure 5 sensors-22-05704-f005:**
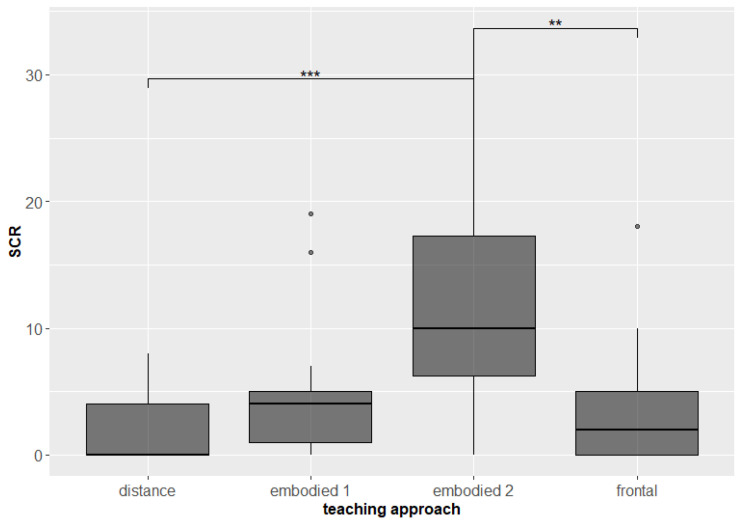
Psychological arousal (measured by SCR) of students during teaching approaches. Significant differences are marked with ** for *p* < 0.01, and *** for *p* < 0.001.

**Figure 6 sensors-22-05704-f006:**
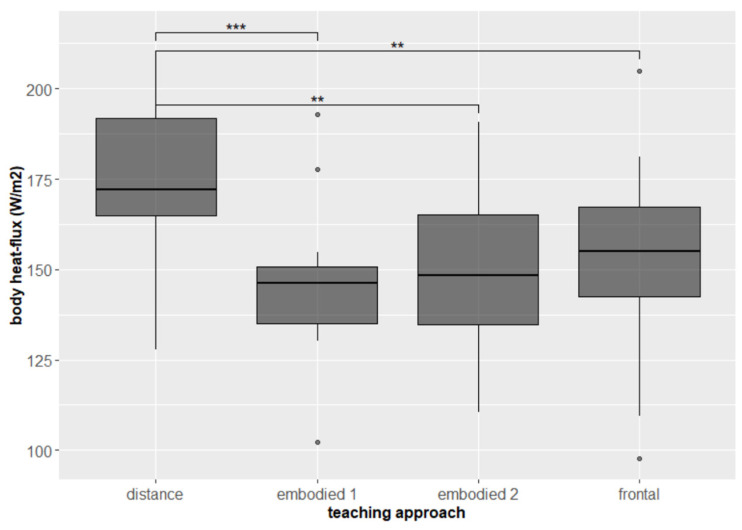
Psychological arousal (measured by HF) of students during teaching approaches. Significant differences are marked with ** for *p* < 0.01, and *** for *p* < 0.001.

**Figure 7 sensors-22-05704-f007:**
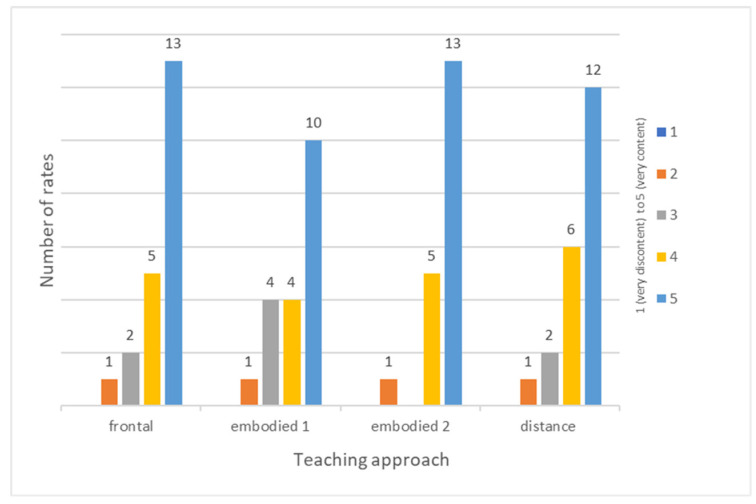
Perception of well-being as described by the students.

**Figure 8 sensors-22-05704-f008:**
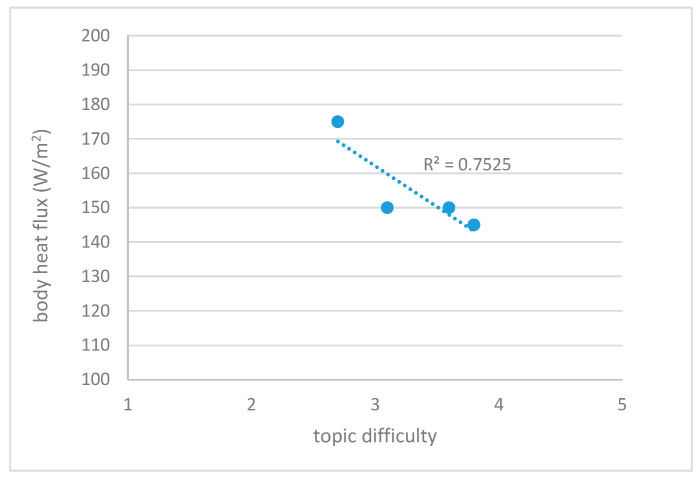
Correlation of physiology and topic difficulty.

**Figure 9 sensors-22-05704-f009:**
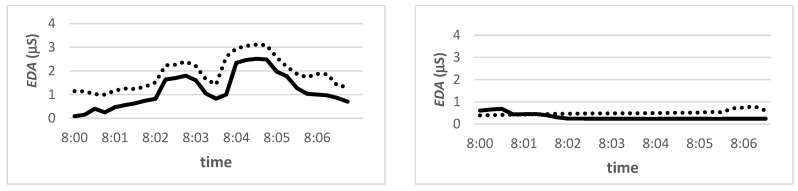
An example of synchronicity of EDA signals of the teacher (solid line) and the average student (dotted line) during embodied teaching. **Left**—synchronized EDA signals (synchronization index 5.0). **Right**—non-synchronized EDA signals (synchronization index 0.2).

**Table 1 sensors-22-05704-t001:** Topics of the four sessions.

Session	Topic	Teaching Approach	Difficulty	Content Description
1	Soil layers and pollution	Frontal (sedentary)	3.6	Presenting selected examples indicating direct and indirect soil pollution (e.g., traffic accidents, road salt). Demonstrating building and testing a water purification filter. Discussing in pairs and working with concrete material in groups. Elaboration of a mind map.
2	States of matter	Embodied	3.8	Demonstrating different states of water and the phase transitions with a demonstration experiment and creative movement in groups. Explaining mass of water in phase transitions.
3	Water cycle	Embodied	3.1	Presenting the water cycle in groups using creative movement with background music. Completing the water cycle chart.
4	Water on Earth	Distance (sedentary)	2.7	Watching a documentary film about surface water. Independently conducting an experiment. Discussing dilemmas and questions, including visuals and mime. Preparing the tabular presentation.

**Table 2 sensors-22-05704-t002:** Percentage of achieved points on knowledge tests and knowledge gain expressed by g-factor (n = number of answers; M = arithmetic mean; SD = standard deviation).

Topic Description	Number of Achieved Points (%) on Pre-Test			Number of Achieved Points (%) on Post-Test 1			g_1_		Number of Achieved Points (%) on Post-Test 2			g_2_	
	n	M	SD	n	M	SD	M	SD	n	M	SD	M	SD
Soil layers and pollution (frontal)	23	50.3	21.0	21	64.6	17.1	0.25	0.37	24	61.5	15.6	−0.21	0.7
States of matter (embodied 1)	20	29.6	19.8	19	55.0	26.1	0.40	0.31	24	42.0	19.1	−0.35	0.6
Water cycle (embodied 2)	20	34.1	14.6	19	67.5	26.7	0.57	0.30	24	49.3	30.9	−0.48	0.9
Water on Earth (distance)	20	40.5	16.3	25	67.6	21.8	0.44	0.33	22	59.6	23.3	−0.44	1.1

## Data Availability

Data supporting reported results can be delivered upon request.

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
