# Peer review of "Are Psychophysiological Wearables Suitable for Comparing Pedagogical Teaching Approaches?"

_sensors, 2022, doi:10.3390/s22155704_

Round 1
Reviewer 1 Report
The paper demonstrates an interesting application of wearable sensors within the classroom. There are several issues I have with the paper, and they need to be addressed before the paper is resubmitted for review.
After reading the paper, I’m not certain as to the overall research subject within the paper. Do the authors suggest that wearable sensors should be used in future classroom work? Or was the justification of the research around whether sensors could be used to prove a point that more activity could lead to better synchronicity between students and teachers? And the authors seem to rely heavily on the statistical analysis and less on describing or discussing how real-world effects could have an effect on the results. So the paper needs some rework.
Although the authors showed the “body bug” wearable sensor in Figure 2, it would be beneficial to add a close-up image of the sensor so that the reader can grasp what the sensor looks like.
The authors describe in the paper how they measured “skin temperature, electrodermal activity (EDA, skin conductance level (SCL) and skin conductance response (SCR)) and heat-flux of the upper-arm, and energy expenditure and estimated physical activity of the student (metabolic equivalent of task (MET)). The authors haven’t mentioned in the paper any statistics or online data on the accuracy ad reliability of this sensor. For example, what level of accuracy and reliability was achievable with this device? And has this device been used in many previous studies? What have other previous research studies found from their assessment of this sensor? Therefore, the authors need to add a relevant section to describe the accuracy and reliability of this sensor so that the reader can gain an understanding on how much the final results can be accepted in terms of accuracy and error rate.
The authors stated in line 196 that “To estimate the physiological synchronisation between two signals, Pearson correlations were calculated in 1-min time windows”. This statement needs further explanation. Why was a 1-minute window chosen for statistical analysis? What was the original sample rate of the sensor, and how was the data summarised to 1 sample per minute? Were the authors not concerned about losing granularity within the data when they downsampled? The authors should add a section in the paper to discuss these points.
Figure 3 does not add anything to the paper and is just a filler. I suggest that the authors should remove it and instead use this valuable space for something more meaningful.
The authors state in line 170 that “All four sessions were video recorded”. How were the sessions recorded at home? Please add a short description to explain how this happened when guidance was not available in the home environment. And was a parent required for the remote session?
Table 1 shows a general description of the 4 sessions and their general topic selections. As a third-level academic who is unfamiliar with the content of these lessons (and a lot of the future readers of this paper may be PhD / post-doc researchers), it would be beneficial to add another column to this table to show in more detail what was taught and demonstrated within each of these sessions. And it will also benefit the reader if the authors add a general discussion on how they envisaged that the students would physically and emotionally react to the stimulus provided by the teachers in each session. I suggest that the authors should add additional information to this table to describe and discuss these comments.
The data shown in Figs 4 and 5 need some more explanation for the “distance” learning tasks. What instructions did the authors provide to students for physical activities for their remote lessons? Did the remote lessons contain the same level of physical activities as the in-school activities? If the students needed to watch videos or to take part in activities by watching a live online session, then we would expect them to be more sedentary throughout the session because of the need to sit and watch instructions provided on-screen. The authors need to add more detail on firstly what limitations were expected with the online sessions compared to classroom sessions, and then they need to put any limitations into context within their discussions for this group.
The authors comment in line 316-317 that “Students were more focused on the teacher in frontal and distance teaching, whereas when they worked in groups during embodied learning, they were focused more on each other and paid less attention to the teacher”. It is unclear what activities the students were completing when they worked individually and in groups, and this may become clearer when the authors update Table 1 as requested in my earlier comment. However, would the authors not expect the students to focus more on the teacher when working individually, and then focus more on their peers in group work? I suggest that the authors need to comment further on this point. Our knowledge of how students react to their teacher within the classroom environment needs to be taken into consideration when any such comments are made.
Figure 10 suggests that there was no synchronicity between the teacher and the average student. And in fact, the balance of positive and negative correlations between teacher and student data at the beginning and end of the session seem to suggest that a false positive effect of 0.69 has been generated. There is no synchronicity between the data at any point in the chart. Therefore, the logic of the authors needs further discussion and clarification. The reader needs to have further explanation on how the authors arrived at the 0.69 positive effect when in fact it is obvious there is no correlation between both data. I suggest that the authors need to either explain their logic, or alternatively they need to amend their evaluation of the correlation.
What do the authors mean by the term “pinch of results” in line 337? Please rephrase to clarify what you mean.
The authors comment in lines 373-375 that “The results show that the more difficult the topic was, the lower the HF levels and average skin temperatures were, and conversely, the easier the topic, the higher the ST and HF levels were”. But the difficulty level has not been described throughout the paper. The difficulty level of each session needs to be clearly explained, and a justification on why some sessions are more difficult than others also need to be described. For example, how can the authors deduce that the distance learning session was more complicated than the other sessions? And were there other possible factors that could have had an emotional response with the students? How can the authors be sure that the issues caused by the teacher’s unstable internet connection did not cause a higher physiological response? And how skilled were the students in connecting to the remote session? Could it be they were stressed by the experience of remote learning, and not too much with the session content? The authors need to add a detailed discussion on their thoughts on unrelated physiological factors such as internet connectivity and complexity of connecting remotely to the session could have on the data, and whether these factors should be considered and ultimately whether they had an adverse effect on the final data.
Structural comments
The first two paragraphs in the introduction section are not in the same font as the other sections. And this error persists in section 2, and throughout the paper. Please amend the paper accordingly.
Line 42 – change to “…in which the human sensory system is activated by delivering new…”
Line 142 – issues with “ ‘ ‘ “ shown in the sentence. Please resolve this issue.
Multiple sentences could be explained better. Please carefully re-read the entire paper, and perhaps get a native English speaker to proof-read it.
Author Response
Dear Editor,
We hereby attach our responses to the reviewers' concerns, comments, and observations. The rest of the document is color-coded as follows: black text - reviewers' comments, red text - authors' response. In the revised paper, the changes are marked in red.
We thank you for your consideration.
Yours sincerely,
Gregor Geršak
REVIEWER 1
The paper demonstrates an interesting application of wearable sensors within the classroom. There are several issues I have with the paper, and they need to be addressed before the paper is resubmitted for review.
After reading the paper, I’m not certain as to the overall research subject within the paper. Do the authors suggest that wearable sensors should be used in future classroom work? Or was the justification of the research around whether sensors could be used to prove a point that more activity could lead to better synchronicity between students and teachers? And the authors seem to rely heavily on the statistical analysis and less on describing or discussing how real-world effects could have an effect on the results. So the paper needs some rework.
We thank you for your careful reading of the manuscript. The main idea of the paper is that it is worthwhile to: i) use wearables in the classroom and ii) use wearables to conduct research, e.g., to optimize teaching to achieve learning objectives. Another interesting finding is that wearables can be used to monitor synchronization between students and teachers to achieve more efficient learning (as evidenced by better knowledge growth on knowledge tests) and to better understand the complex process of learning.
Based on the reviewer’s comment, we have added and/or changed a number of sentences throughout the paper for the reader's better understanding (marked in red).
Although the authors showed the “body bug” wearable sensor in Figure 2, it would be beneficial to add a close-up image of the sensor so that the reader can grasp what the sensor looks like.
We have added a close-up image of the sensor (Figure 2).
The authors describe in the paper how they measured “skin temperature, electrodermal activity (EDA, skin conductance level (SCL) and skin conductance response (SCR)) and heat-flux of the upper-arm, and energy expenditure and estimated physical activity of the student (metabolic equivalent of task (MET)). The authors haven’t mentioned in the paper any statistics or online data on the accuracy ad reliability of this sensor. For example, what level of accuracy and reliability was achievable with this device?
The metrological characteristics of the sensor were evaluated in many studies [1–9]; the accuracy (i.e. measuring uncertainty) of energy expenditure measurements has been estimated to 10 %, skin conductance 1 %, skin temperature 5 % and body heat-flux to approximately 20 %. We have added this information to the 2.2 Instrumentation section.
- Teller, A. A platform for wearable physiological computing. Interact. Comput. 2004, 16, 917–937.
- Sharif, M.M.; Bahammam, A.S. Sleep estimation using BodyMedia’s SenseWearTM armband in patients with obstructive sleep apnea. Ann. Thorac. Med. 2013, 8, 53–7.
- Geršak, G.; Drnovšek, J. Sensewear body monitor in psychophysiological measurements. In Proceedings of the IFMBE Proceedings; 2016; Vol. 57.
- Welk, G.J.; McClain, J.J.; Eisenmann, J.C.; Wickel, E.E. Field validation of the MTI Actigraph and BodyMedia armband monitor using the IDEEA monitor. Obesity (Silver Spring). 2007, 15, 918–28.
- Papazoglou, D.; Augello, G.; Tagliaferri, M.; Savia, G.; Marzullo, P.; Maltezos, E.; Liuzzi, A. Evaluation of a multisensor armband in estimating energy expenditure in obese individuals. Obesity 2006, 14, 2217–2223.
- Liden, C.B.; Wolowicz, M.; Ed, M.; Des, J.S.M.; Teller, A.; Ph, D.; Vishnubhatla, S.; Ee, M.S.; Pelletier, R.; Lc, M.S.; et al. Accuracy and Reliability of the SenseWear TM Armband as an Energy Expenditure Assessment Device. 2002, 1–15.
- Calabro, M.A.; Lee, J.M.; Saint-Maurice, P.F.; Yoo, H.; Welk, G.J. Validity of physical activity monitors for assessing lower intensity activity in adults. Int. J. Behav. Nutr. Phys. Act. 2014, 11.
- Ogorevc, J.; Geršak, G.; Novak, D.; Drnovšek, J. Metrological evaluation of skin conductance measurements. Meas. J. Int. Meas. Confed. 2013, 46, 2993–3001.
- Powell, C.; Carson, B.P.; Dowd, K.P.; Donnelly, A.E. The accuracy of the SenseWear Pro3 and the activPAL3 Micro devices for measurement of energy expenditure. Physiol. Meas. 2016, 37, 1715–1727.
And has this device been used in many previous studies? What have other previous research studies found from their assessment of this sensor? Therefore, the authors need to add a relevant section to describe the accuracy and reliability of this sensor so that the reader can gain an understanding on how much the final results can be accepted in terms of accuracy and error rate.
The reliability and accuracy of the sensor used was thoroughly evaluated in many studies [1–9]. We have added this information to the 2.2 Instrumentation section.
- Teller, A. A platform for wearable physiological computing. Interact. Comput. 2004, 16, 917–937.
- Sharif, M.M.; Bahammam, A.S. Sleep estimation using BodyMedia’s SenseWearTM armband in patients with obstructive sleep apnea. Ann. Thorac. Med. 2013, 8, 53–7.
- Geršak, G.; Drnovšek, J. Sensewear body monitor in psychophysiological measurements. In Proceedings of the IFMBE Proceedings; 2016; Vol. 57.
- Welk, G.J.; McClain, J.J.; Eisenmann, J.C.; Wickel, E.E. Field validation of the MTI Actigraph and BodyMedia armband monitor using the IDEEA monitor. Obesity (Silver Spring). 2007, 15, 918–28.
- Papazoglou, D.; Augello, G.; Tagliaferri, M.; Savia, G.; Marzullo, P.; Maltezos, E.; Liuzzi, A. Evaluation of a multisensor armband in estimating energy expenditure in obese individuals. Obesity 2006, 14, 2217–2223.
- Liden, C.B.; Wolowicz, M.; Ed, M.; Des, J.S.M.; Teller, A.; Ph, D.; Vishnubhatla, S.; Ee, M.S.; Pelletier, R.; Lc, M.S.; et al. Accuracy and Reliability of the SenseWear TM Armband as an Energy Expenditure Assessment Device. 2002, 1–15.
- Calabro, M.A.; Lee, J.M.; Saint-Maurice, P.F.; Yoo, H.; Welk, G.J. Validity of physical activity monitors for assessing lower intensity activity in adults. Int. J. Behav. Nutr. Phys. Act. 2014, 11.
- Ogorevc, J.; Geršak, G.; Novak, D.; Drnovšek, J. Metrological evaluation of skin conductance measurements. Meas. J. Int. Meas. Confed. 2013, 46, 2993–3001.
- Powell, C.; Carson, B.P.; Dowd, K.P.; Donnelly, A.E. The accuracy of the SenseWear Pro3 and the activPAL3 Micro devices for measurement of energy expenditure. Physiol. Meas. 2016, 37, 1715–1727.
The authors stated in line 196 that “To estimate the physiological synchronisation between two signals, Pearson correlations were calculated in 1-min time windows”. This statement needs further explanation. Why was a 1-minute window chosen for statistical analysis? What was the original sample rate of the sensor, and how was the data summarised to 1 sample per minute? Were the authors not concerned about losing granularity within the data when they downsampled? The authors should add a section in the paper to discuss these points.
The rather low sampling frequency of the wearable which was only 4 samples per minute resulted in selecting the length of the time window for correlation calculations of 1 minute. Using the correlation coefficients a synchronicity index was calculated as the ratio of positive coefficients and negative coefficients as suggested in [10,11]. Thus, positive value of the synchronicity index represented a stronger link of the selected signals[10].
A paragraph on this issue was added to the text and included in limitation of the study.
- Marci, C.D.; Ham, J.; Moran, E.; Orr, S.P. Physiologic correlates of perceived therapist empathy and social-emotional process during psychotherapy. J. Nerv. Ment. Dis. 2007, 195, 103–111.
- Marci, C.D.; Orr, S.P. The effect of emotional distance on psychophysiologic concordance and perceived empathy between patient and interviewer. Appl. Psychophysiol. Biofeedback 2006, 31, 115–128.
Figure 3 does not add anything to the paper and is just a filler. I suggest that the authors should remove it and instead use this valuable space for something more meaningful.
As suggested, we removed the Figure 3.
The authors state in line 170 that “All four sessions were video recorded”. How were the sessions recorded at home? Please add a short description to explain how this happened when guidance was not available in the home environment. And was a parent required for the remote session?
Thank you for the remark. Three lessons took place at school and one at home. The lessons at school were videotaped, whereas for the online lessons the recording in MS Teams application was used (i.e. only faces of the students were visible). In addition, students received special instructions on installation of wearables before the online lessons, even before the sensors were distributed to them. Due to the previous usage in school the students were already familiar with MS Teams application and with sensors, therefore no help from the parents was needed for the remote session (and thus their direct influence avoided).
Table 1 shows a general description of the 4 sessions and their general topic selections. As a third-level academic who is unfamiliar with the content of these lessons (and a lot of the future readers of this paper may be PhD / post-doc researchers), it would be beneficial to add another column to this table to show in more detail what was taught and demonstrated within each of these sessions. And it will also benefit the reader if the authors add a general discussion on how they envisaged that the students would physically and emotionally react to the stimulus provided by the teachers in each session. I suggest that the authors should add additional information to this table to describe and discuss these comments.
We have expanded the Table 1 and added the suggested information. Please note that the science topics lessons were prepared according to the national curricula of the subject and that the detailed analysis of the activities from the lessons is not in the scope of the journal and probably not interesting for the Sensors readers. However, we believe that added description of contents will help to reach a better understanding of the situation in science classroom.
Session |
Topic |
Teaching approach |
Difficulty |
Content description |
1 |
Soil layers and pollution |
Frontal |
3.6 |
Presenting selected examples indicating direct and indirect soil pollution (e.g., traffic accidents, road salt, etc.). Demonstrating building and testing a water purification filter. Discussing in pairs and work with concrete material in groups. Elaboration of a mind map. |
2 |
States of matter |
Embodied |
3.8 |
Demonstrating different states of water and the phase transitions with a demonstration experiment and creative movement in groups. Explaining mass of water in phase transitions. |
3 |
Water cycle |
Embodied |
3.1 |
Presenting the water cycle in groups using creative movement with background music. Completing the water cycle chart. |
4 |
Water on Earth |
Distance |
2.7 |
Watching a documentary film about surface water. Independently conducting an experiment. Discussing dilemmas and questions, including visuals, mime. Preparing the tabular presentation. |
The data shown in Figs 4 and 5 need some more explanation for the “distance” learning tasks. What instructions did the authors provide to students for physical activities for their remote lessons?
The teacher was instructed to design the distance-teaching lesson in such a way that the students would physically move similarly as during the frontal work in school.
This text was included in the paper.
Did the remote lessons contain the same level of physical activities as the in-school activities?
Yes, they did. As shown in Fig 4 and Fig 5 the energy expenditure was similar at frontal and distance lessons (in comparison with the embodied lessons both being mostly sedentary).
If the students needed to watch videos or to take part in activities by watching a live online session, then we would expect them to be more sedentary throughout the session because of the need to sit and watch instructions provided on-screen. The authors need to add more detail on firstly what limitations were expected with the online sessions compared to classroom sessions, and then they need to put any limitations into context within their discussions for this group.
Thank you for this comment. We added some more explanation in the section 2.3 Study protocol. Similar to frontal lessons in school (as described in Table 1) the online teaching was physically similarly active. It was predominantly of sedentary type. The teacher was instructed to design the distance-teaching lesson in such a way that the students would physically move similarly as during the frontal work in school (Fig 4 and Fig 5 are the proofs that it worked). Therefore, the main possible limitations of online lessons as compared to frontal teaching were the quality of the internet connection and distractions at home.
The authors comment in line 316-317 that “Students were more focused on the teacher in frontal and distance teaching, whereas when they worked in groups during embodied learning, they were focused more on each other and paid less attention to the teacher”. It is unclear what activities the students were completing when they worked individually and in groups, and this may become clearer when the authors update Table 1 as requested in my earlier comment. However, would the authors not expect the students to focus more on the teacher when working individually, and then focus more on their peers in group work? I suggest that the authors need to comment further on this point. Our knowledge of how students react to their teacher within the classroom environment needs to be taken into consideration when any such comments are made.
It was expected that students were more focused on the teacher in frontal and distance teaching when working individually and more focused on each other (paying less attention to the teacher) when working in groups during embodied teaching. And this was actually proven using the calculated synchronization indices (Comparing synchronization indices between teacher and students, we found that synchronization index during embodied teaching was 0.69 and 0.61 (for two lessons), during frontal teaching 1.25 and during distance teaching 1.07). The results suggest that synchronization index could be a measure of student focus. For example, lower index. during the embodied teaching can be clarified - the teacher explained the topics while the students were involved in their own creative process of using their body and movements to represent the topics, i.e. students were less focused on the teacher.
Figure 10 suggests that there was no synchronicity between the teacher and the average student. And in fact, the balance of positive and negative correlations between teacher and student data at the beginning and end of the session seem to suggest that a false positive effect of 0.69 has been generated. There is no synchronicity between the data at any point in the chart. Therefore, the logic of the authors needs further discussion and clarification. The reader needs to have further explanation on how the authors arrived at the 0.69 positive effect when in fact it is obvious there is no correlation between both data. I suggest that the authors need to either explain their logic, or alternatively they need to amend their evaluation of the correlation.
We thank the reviewer for the valuable comment. What we wanted was to present the reader difference between synchronized and non-synchronized signals of electrodermal activity. We showed how non-synchronized signals look like and forgot to include an example of well-synchronized signals. We have changed the graphs for the reader to see the difference.
What do the authors mean by the term “pinch of results” in line 337? Please rephrase to clarify what you mean.
We rephrased the line to “Table 2 shows the results related to knowledge gain.”
The authors comment in lines 373-375 that “The results show that the more difficult the topic was, the lower the HF levels and average skin temperatures were, and conversely, the easier the topic, the higher the ST and HF levels were”. But the difficulty level has not been described throughout the paper.
In fact, in section 2.3 Study protocol, the estimating the topic’s difficulty by an expert panel is already described (see below).
“The topics were selected from the Slovenian national curriculum for Science and Technology [79–81]. To explore correlation of physiology with topic difficulty and to control for differences in topics difficulty, prior to this study, an expert panel (six experienced teachers of science) estimated topic difficulty on a 5-point Likert scale, 1 being the easiest and 5 the hardest topic (Table 1).”
The difficulty level of each session needs to be clearly explained, and a justification on why some sessions are more difficult than others also need to be described. For example, how can the authors deduce that the distance learning session was more complicated than the other sessions?
We understand the reviewer’s concerns. We were interested in the difficulty of the lessons conducted in the classroom. Six people from the field of science didactics were given an overview of the lesson plans. Topics were rated on a five-point scale, where 1 meant the easiest lesson and 5 meant the most difficult lesson. According to selected experts in the field of science didactics, the lesson about water on Earth, conducted remotely, was the easiest lesson, and the lesson about states of mater on submicroscopic level was the most difficult.
And were there other possible factors that could have had an emotional response with the students? How can the authors be sure that the issues caused by the teacher’s unstable internet connection did not cause a higher physiological response? And how skilled were the students in connecting to the remote session? Could it be they were stressed by the experience of remote learning, and not too much with the session content? The authors need to add a detailed discussion on their thoughts on unrelated physiological factors such as internet connectivity and complexity of connecting remotely to the session could have on the data, and whether these factors should be considered and ultimately whether they had an adverse effect on the final data.
Other possible influences that could affect the emotional response of the students are a new teacher and the researcher present, but this was kept to a minimum since both had been present in the class several times before and the students already knew them. The class is a living system, so unplanned events such as another teacher arriving in the classroom are possible and not unexpected for the students.
In distance learning, students were already at home during the study. Students did not perceive any noticeable reactions to the teacher's unstable Internet connection. In fact, as visible from the video recordings, some students did not even notice it. It should be emphasized that from the observation of the students' reactions and from the answers in the students' well-being questionnaires, it can be concluded that there were no significant changes in their reactions.
It is an interesting question what they focused more on during the distance learning: on the distance learning itself or on the learning content. From the results of the knowledge test, we can conclude that the focus on the learning content was noticeable. Of course, distance learning and data collection in this way has its pitfalls, but because of the trust we built with the students, we believe we greatly reduced the negative impact as they made an effort to participate similar to what they did in school.
The paper has been modified to include some of these statements.
Structural comments
The first two paragraphs in the introduction section are not in the same font as the other sections. And this error persists in section 2, and throughout the paper. Please amend the paper accordingly.
We amended the paragraphs format throughout the paper.
Line 42 – change to “…in which the human sensory system is activated by delivering new…”
We corrected the sentence.
Line 142 – issues with “ ‘ ‘ “ shown in the sentence. Please resolve this issue.
We corrected the sentence.
Multiple sentences could be explained better. Please carefully re-read the entire paper, and perhaps get a native English speaker to proof-read it.
We had the work proofread by a native speaker.

Reviewer 2 Report
The author has presented an interesting and detailed study related to the the wearable sensors to be utilized in pedagogical teaching approaches. The study is well co-related with the literature and significant amount of references are quoted. I have the following suggestions to improve the quality of the paper.
1) At the end of the Introduction section, a paragraph should be added which presents the structure of the paper. This will help the reader to drive through the paper. Otherwise, its an abrupt change from Introduction to the next section.
2) What is the repeatability of the experiment? And the percentage of error?
3) Can this study or the analysis obtained from this study be generally applied to other countries? Or it is specific country based study?
4) If possible, can you mention the cost of the sensor in the paper and add some zoom image of the device?
5) I suggest avoid using references in the Conclusion section.
Author Response
REVIEWER 2
The author has presented an interesting and detailed study related to the the wearable sensors to be utilized in pedagogical teaching approaches. The study is well co-related with the literature and significant amount of references are quoted. I have the following suggestions to improve the quality of the paper.
1) At the end of the Introduction section, a paragraph should be added which presents the structure of the paper. This will help the reader to drive through the paper. Otherwise, its an abrupt change from Introduction to the next section.
Thank you for the comment. We added the following paragraph to the Introduction:
The paper is structured as follows: After the Introduction, in which an extensive literature review is conducted to identify the role of wearable devices in educational science and to present different teaching approaches, the following Materials and Methods section details participants, measurement instruments, study protocol, and data processing. The following Results section consists of the main findings and their interpretation and is followed by a comprehensive Discussion. The paper is rounded off by a brief Conclusion.
2) What is the repeatability of the experiment? And the percentage of error?
The metrological properties of the sensor have been evaluated in numerous studies [1–9]; the accuracy (i.e., measurement uncertainty) of energy expenditure measurements has been estimated to be 10%, skin conductance 1%, skin temperature 5%, and body heat flux approximately 20%. We have included this information in Section 2.2 Instrumentation.
In our experience, the wearables used in this study are quite repeatable. The repeatability of measurements, albeit only for skin conductance and skin temperature, was studied in detail in one of our previous papers [8].
- Teller, A. A platform for wearable physiological computing. Interact. Comput. 2004, 16, 917–937.
- Sharif, M.M.; Bahammam, A.S. Sleep estimation using BodyMedia’s SenseWearTM armband in patients with obstructive sleep apnea. Ann. Thorac. Med. 2013, 8, 53–7.
- Geršak, G.; Drnovšek, J. Sensewear body monitor in psychophysiological measurements. In Proceedings of the IFMBE Proceedings; 2016; Vol. 57.
- Welk, G.J.; McClain, J.J.; Eisenmann, J.C.; Wickel, E.E. Field validation of the MTI Actigraph and BodyMedia armband monitor using the IDEEA monitor. Obesity (Silver Spring). 2007, 15, 918–28.
- Papazoglou, D.; Augello, G.; Tagliaferri, M.; Savia, G.; Marzullo, P.; Maltezos, E.; Liuzzi, A. Evaluation of a multisensor armband in estimating energy expenditure in obese individuals. Obesity 2006, 14, 2217–2223.
- Liden, C.B.; Wolowicz, M.; Ed, M.; Des, J.S.M.; Teller, A.; Ph, D.; Vishnubhatla, S.; Ee, M.S.; Pelletier, R.; Lc, M.S.; et al. Accuracy and Reliability of the SenseWear TM Armband as an Energy Expenditure Assessment Device. 2002, 1–15.
- Calabro, M.A.; Lee, J.M.; Saint-Maurice, P.F.; Yoo, H.; Welk, G.J. Validity of physical activity monitors for assessing lower intensity activity in adults. Int. J. Behav. Nutr. Phys. Act. 2014, 11.
- Ogorevc, J.; Geršak, G.; Novak, D.; Drnovšek, J. Metrological evaluation of skin conductance measurements. Meas. J. Int. Meas. Confed. 2013, 46, 2993–3001.
- Powell, C.; Carson, B.P.; Dowd, K.P.; Donnelly, A.E. The accuracy of the SenseWear Pro3 and the activPAL3 Micro devices for measurement of energy expenditure. Physiol. Meas. 2016, 37, 1715–1727.
On the other hand, the repeatability of the study presents a greater challenge. It would be possible to repeat the study in the classroom, but since the relevant circumstances change and the classroom is a living system (including the season, the time of day, the mood of the students, the teacher, etc.), we should expect somewhat different results. Nevertheless, the main goal of the article-are wearables useful for studying the dynamics in the classroom-is relatively straightforward, i.e., from our results we can conclude that wearables can be useful for studying the complex learning process in the classroom.
3) Can this study or the analysis obtained from this study be generally applied to other countries? Or it is specific country based study?
Due to the size of the sample, the results cannot be generalized to the general population, but we expect similar results in other countries, as science curricula elsewhere are also similar to those in Slovenia. Much also depends on the preparation of the teacher. Since usually only enthusiastic teachers are willing to cooperate in this type of research, we expect similar results.
We added this text in the Discussion section.
4) If possible, can you mention the cost of the sensor in the paper and add some zoom image of the device?
The BodyBugg device is no longer manufactured, so we see no need to include it in the paper. The price ranged from 300 USD (latest version) to 1000 USD (early versions). A photo of the wearable has been included (Figure 2).
5) I suggest avoid using references in the Conclusion section.
We removed the references from the Conclusion.

Round 2
Reviewer 1 Report
The authors have dealt with most of my comments from the previous round of reviews and the paper is much better now. I've some minor comments that the authors should work on before submitting the paper.
Bottom of page 3 - fill up the white space.
The authors could amalgamate Figs 2 and 3 by placing both side-by-side into one figure.
Table 1 should not be split across two pages.
Figures 4-8 need their sizes increased to make them easier to read.
Figure 10 - y-axis values need to be shown, and relevant label with unit of measurement should also be shown.
Author Response
REVISED PAPER – second revision
Are psychophysiological wearables suitable for comparing pedagogical teaching approaches?
Manuscript ID: sensors-1810730
Dear Editor,
We hereby attach our responses to the reviewer's comments. Again the document is color-coded as follows: black text - reviewers' comments, red text - authors' response. In the revised paper, the changes are marked in red.
Yours sincerely,
Gregor Geršak
REVIEWER 1
The authors have dealt with most of my comments from the previous round of reviews and the paper is much better now. I've some minor comments that the authors should work on before submitting the paper.
Bottom of page 3 - fill up the white space.
We deleted the empty lines.
The authors could amalgamate Figs 2 and 3 by placing both side-by-side into one figure.
We did as suggested.
Table 1 should not be split across two pages.
We did as suggested.
Figures 4-8 need their sizes increased to make them easier to read.
We resized the figures.
Figure 10 - y-axis values need to be shown, and relevant label with unit of measurement should also be shown.
We thought that the absolute values of skin conductance are not very important for the reader to understand the fundamental difference between synchronized and non-synchronized signals - namely, we are comparing the dynamics of the teacher's physiology with the dynamics of the of the students' physiology. Now we have added the units.

This manuscript is a resubmission of an earlier submission. The following is a list of the peer review reports and author responses from that submission.
Round 1
Reviewer 1 Report
This article examines the usefulness of wearable devices to analyze and compare learning experiences.
Although the use of wearable devices in education can be considered a novel topic, in the humble opinion of this reviewer, the article has serious flaws that prevents it to make a significant contribution.
In educational research, when different learning approaches or activities are compared, the two most important aspects to be analyzed are learning gains and student motivation. In the comparison presented in this article, these aspects are overlooked. Some physiological data was collected, but there was no solid link between these data and learning gains and/or student motivation.
Moreover, the research design is not solid. When comparing different learning approaches, there should be only one independent variable in order to obtain reliable results. In the study presented in this article, besides the learning approach employed, there are more independent variables (topic, teacher, etc.). For future studies, I suggest authors to use solid research designs such as randomized controlled trial, or at least a research design with control and experimental groups and only one independent variable.
Another major limitation of this work is the small sample size.
Taking into account all the issues previously described, it can be stated that the findings that this article demonstrates that can be obtained by using wearable devices are not very useful because they are not reliable enough and do not provide a clear understanding of the learning approaches in terms of instructional effectiveness.
Reviewer 2 Report
Title: The measurement present in the abstract is not the objective present in the title: “ are wearables suitable for comparison?”
Perhaps a title could be similar to the following: “Comparing three pedagogical approaches for teaching Science using wearables at the elementary school – ….”
Are the approaches didactical, learning or pedagogical?
Abstract = Objectives are not clear: how can the three proposed approaches be measured in terms of a “psychological arousal” and “perceived well-being of the students”? How is synchronization measured?
Revision of English: “ This study demonstrates a use of wearables in the classroom during science lessons.” (it is not clear)
Among the keywords there is “ kinaesthetic teaching”, but this is not present nor in the title or in the abstract.
In the title of the work we find three theoretical approaches, but there is no explanation in the introduction, in particular there is no reference to Constructivism and Constructionism about the connection between learning and action). In fact, in the paper we can read: “ One of the challenges of education today from kindergarten to university is to reduce students' sedentary time and incorporate more physical activities into the curriculum in order to improve concepts such as learning outcomes, student well-being, learning effectiveness, long-term retention, etc.”
Maybe the following reference could be changed into one more appropriate: (is the sitting explored or the learning through movement-action?) Routen, A.C. Should our children be sitting comfortably in school? BMJ 2011, 343, d4273.
It is supposed that every reader knows the Embodied teaching, but it is necessary to explain it (briefly) in order to fully understand the aim of the paper.
One of the goals of this study was to create a tool for estimating the physiological synchronicity of persons involved in the process of teaching and learning = However in the introduction the authors should explain the usefulness of this measurement.
One of the tools that help researchers to discover processes and activities within learning (and teaching) when placed in a real environment are wearable devices: another one is the visual monitoring of the gaze (some reference could be added).
The physiological connection between two persons was extensively studied in other environments.= why? Results? References?
in the fields of education, philosophy, psychology, psychotherapy, neuroscience etc [29–32]. = studies already present in the field of education should be briefly cited and explained.
Line 65-66 = Note, that for the purpose of this paper by physiological synchronisation we mean synchronisation of the electrodermal activity (EDA). = this part should be moved into the methodology and explained
Line 72 =the physiological connection between the two= it is not clear if also the teacher’s physiology is measured
Subjects = Line 75 and Figure 1: In the figure it is understandable if the same students participated to different approaches. “25 eleven-year old fifth graders “ are not a large sample and there is not a control group. Maybe it is only a wrong figure: in this case, what does “ 17-15-18-18” mean?
Line 84-85 = “of total 25 students only data of 21, 19, 22, and 21 students could be processed within four sessions”: are subject only 25? Or has a number been assigned to each student and are these the assigned numbers ?
“A questionnaire for es timation of emotional feeling of the student during learning”= During? when? Half lesson?
Since there are few subjects, in the title it could be added “a pilot study”
Figure 2: why are the students 18?
In methodology: how many hours of teaching? How many days?
Something present in the Discussion part seems more appropriate in the “Results” section.
“Results” from questionnaires should be in a different section.
In conclusion, theoretical part should be improved, and the methodological part should be more complete and precise. The reading of results should be more understandable.
Reviewer 3 Report
The current study used a wearable device as an objective tool to compare three different pedagogical teaching approaches that of classical frontal teaching, holistic teaching based on embodied cognition, and distance teaching approach. These approaches were compared in terms of their impact on students’ psychological arousal and perceived well-being. In addition, an attempt was made to determine the physiological synchronisation between teacher and students during the lecture.
On the whole, the research conducted, and the discussion of the evidence are good. However, there are some issues that the authors must handle.
The authors need to correct some English grammar mistakes and the meaning of some phrases e.g., ‘’The approaches were compared in terms of their impact on students, i.e. psychological arousal and perceived well-being of the students. ‘’ --- suggestion ’The approaches were compared in terms of their impact on students’ psychological arousal and perceived well-being, etc.’’
Align to the center the tables.
Subsections 3.3 and 3.4 are too short. Consider merging them, if possible or extend them.
Also, the authors say: ‘’ One of the goals of this study was to create a tool for estimating….’. Is this a tool that you discussed in your article or a kind of framework or something else?
Also, you say: ‘’The aim of this paper is to present a study in a real-world setting.’’ – Is this the only aim of your study?
Which are the main differences between the current research with those of [25-27], from the same authors(some of them)? The works [25-27] were also used wearable devices to study the activity of the children in a classroom.
Actually, I am not sure about the main topic of your article. For instance, the title says, ‘’…… are wearables suitable for comparison’’. You do not compare different wearable devices. Why plural? Am I right? The same is referred in your document. Please explain this point.
The title of 2.1 must be better Participants.
The title of 2.3 should be clearer. Procedure for what? Learning procedure?
Author Response
Reviewer #3 - Comments and Suggestions for Authors
The current study used a wearable device as an objective tool to compare three different pedagogical teaching approaches that of classical frontal teaching, holistic teaching based on embodied cognition, and distance teaching approach. These approaches were compared in terms of their impact on students’ psychological arousal and perceived well-being. In addition, an attempt was made to determine the physiological synchronisation between teacher and students during the lecture.
On the whole, the research conducted, and the discussion of the evidence are good. However, there are some issues that the authors must handle.
The authors need to correct some English grammar mistakes and the meaning of some phrases e.g., ‘’The approaches were compared in terms of their impact on students, i.e. psychological arousal and perceived well-being of the students. ‘’ --- suggestion ’The approaches were compared in terms of their impact on students’ psychological arousal and perceived well-being, etc.’’
We thank the reviewer for this comment - the text was changed as suggested.
Align to the center the tables.
The table was aligned to the centre.
Subsections 3.3 and 3.4 are too short. Consider merging them, if possible or extend them.
The sections were rearranged – new section 3.1 was extended and the section 3.2 is now devoted solely to physiological synchronisation.
Also, the authors say: ‘’ One of the goals of this study was to create a tool for estimating….’. Is this a tool that you discussed in your article or a kind of framework or something else?
For establishing a measure/level of physiological synchronicity of two individuals, we defined a synchronisation index (in the third paragraph in 2.4 Data analysis in the Methods section). This index is then used in the Results section as a quantitative measure of physiological signal linkage.
Also, you say: ‘’The aim of this paper is to present a study in a real-world setting.’’ – Is this the only aim of your study?
We thank the reviewer for this comment - the text was changed to “One of the main goals of this paper is to present an educational research study using wearable devices in a real-world setting.”
Which are the main differences between the current research with those of [25-27], from the same authors(some of them)? The works [25-27] were also used wearable devices to study the activity of the children in a classroom.
In the study, published in 2020 (Geršak, V et al. Use of wearable devices to study activity of children in classroom; Case study — Learning geometry using movement. Comput. Commun. 2020, 150.) the effects of embodied teaching approach were monitored with a large sample of 104 younger pupils (second-graders) divided into two groups - the control group which was taught classically and the experimental group which was taught by means of movement-based teaching approach (embodied teaching). The topics was not science as in the proposed manuscript, but mathematics and geometry. There was no distance teaching as a teaching approach. The study [25] was indeed a starting point for the research design of the current study, but the only important similarity is the use of the same type of wearable device. Other references (26 and 27) have no relevant similarities with the current study.
Actually, I am not sure about the main topic of your article. For instance, the title says, ‘’…… are wearables suitable for comparison’’. You do not compare different wearable devices. Why plural? Am I right? The same is referred in your document. Please explain this point.
The question, which this paper is primarily dealing with, is not WHICH wearable is suitable for comparison of different teaching approaches, i.e. we are not presenting features/properties of wearables. Instead, we are describing a study in which we research whether the wearables (a wearable is probably grammatically more suitable English naming, although wearables is usually a term encompassing portable measuring devices) can be regarded as a suitable tool in educational research. We are a bit reluctant about the singular version “Is a wearable device suitable for comparing pedagogical teaching approaches?”, so we decided to leave wearables in plural in the title.
The title of 2.1 must be better Participants.
We thank the reviewer for this comment - the title was changed to Participants.
The title of 2.3 should be clearer. Procedure for what? Learning procedure?
We changed the section title to »Study protocol” to better reflect its content (i.e. the description of the method/protocol according to which the study was conducted).

Round 2
Reviewer 1 Report
This article examines the usefulness of wearable devices to analyze and compare learning experiences.
The main concerns raised in my previous review has not been succesfully adresssed in the revised version of the manuscript. Therefore, in the humble opinion of this reviewer, the article still has serious flaws that prevents it to make a significant contribution.
Author Response
Our responses to reviewers' comments can be found below. We believe we have addressed all of his/her concerns, and due to the lack of specificity, we find it difficult to "successfully address them."
Reviewer #1 - Comments and Suggestions for Authors
This article examines the usefulness of wearable devices to analyze and compare learning experiences. Although the use of wearable devices in education can be considered a novel topic, in the humble opinion of this reviewer, the article has serious flaws that prevents it to make a significant contribution.
In educational research, when different learning approaches or activities are compared, the two most important aspects to be analyzed are learning gains and student motivation. In the comparison presented in this article, these aspects are overlooked. Some physiological data was collected, but there was no solid link between these data and learning gains and/or student motivation.
We thank the reviewer for this comment. Of course the obvious effects of every research in education when dealing with different learning approaches would be the level of newly gained knowledge, students motivation and similar phenomena. The main objective of this paper is on the other hand different – it is focused on novelties extremely useful for other sciences involved in educational research in schools – physiology, biomedical engineering, kinematics, biomechanics, ergonomics, HCI, psychophysiology etc. The paper questions the usefulness of (small, non-expensive, simple, relatively low-tech) wearable devices in classrooms and within educational research. Therefore, this paper, sent to the journal Sensors (and intended for Sensors (engineering) readers) is representing the technical side of the research.
Additionally, it should be noted, that the same authors authored another paper resulting from this research. This second paper, which is currently in the process of publishing in a specialised educational science journal, is focused entirely on the comparison of three pedagogical approaches, methodology and the results in terms of pedagogical measures of our study (G factors, short and long-term retention, success of special need students etc). The primary readers of the second paper are educational scientists.
Moreover, the research design is not solid. When comparing different learning approaches, there should be only one independent variable in order to obtain reliable results. In the study presented in this article, besides the learning approach employed, there are more independent variables (topic, teacher, etc.). For future studies, I suggest authors to use solid research designs such as randomized controlled trial, or at least a research design with control and experimental groups and only one independent variable.
We thank the reviewer for this important comment. The comment proves hers/his experience in in-situ studies such as the one described in this paper and points out that the reviewer is well aware of practical ergonomic, logistical, ethical and other practical issues in similar studies. The presented study is considered as a pilot research for a larger future study, which will indeed be composed of several studies, all with control and experimental groups. For the purpose of this paper, we were focused on relative differences of students’ physiology when subject to different types of teaching and finding of possible correlations of physiology and students well being and not on validation or evaluation of different teaching approaches.
Another major limitation of this work is the small sample size.
We thank for the comment. In Discussion section there is a clear statement stressing this limitation “Apart the limited number of subjects, the inherent limitation of the study is the mental, cognitive, as well as environmental complexity of a classroom.“ We believe that all the readers of Sensors, active in the field of wearable devices used in real-life outside laboratory conditions will appreciate that the described study is merely a (small) foundation stone in the rather giant block of how technology can contribute to a better understanding, improving or managing extremely complex activities in a classroom during teaching and learning process. For an indeed more adequate description, we added the adjective “pilot” to the word study (research) in several parts of the manuscript.
Taking into account all the issues previously described, it can be stated that the findings that this article demonstrates that can be obtained by using wearable devices are not very useful because they are not reliable enough and do not provide a clear understanding of the learning approaches in terms of instructional effectiveness.
We do not agree with the reviewer. This, actually, is precisely the reason why we think this study should be published in Sensors. There is a (growing) number of researchers publishing in Sensors, which are very interested in the benefits of using wearables (and not only classical biomechanical ones, but also the novel ones, focused on the psychophysiological state of the wearer). They need to get new ideas, see new applications of old methodologies in new settings and they need to read about rather successful cases. The statement that the wearable devices are not reliable enough for a meaningful research, should maybe be reassessed, e.g. consider a large corpus of papers only on energy expenditure with respect of obesity in nowadays schools.
Reviewer 2 Report
Revise English in line 40-41. I suggest to revise minor mistakes throughout all the paper
Author Response
The text »Based on these findings and in contrast to classical sedentary ex-cathedra frontal teaching, to so-called embodied (kinaesthetic) teaching approach was developed.« was simplified to »Based on these findings the so-called embodied (kinaesthetic) teaching approach was developed.«